# sFLT1, PlGF, the sFLT1/PlGF Ratio and Their Association with Pre-Eclampsia in Twin Pregnancies—A Review of the Literature

**DOI:** 10.3390/medicina59071232

**Published:** 2023-06-30

**Authors:** Ioakeim Sapantzoglou, Angeliki Rouvali, Antonios Koutras, Maria Ioanna Chatziioannou, Ioannis Prokopakis, Zacharias Fasoulakis, Eleftherios Zachariou, Athanasios Douligeris, Anastasia Mortaki, Paraskevas Perros, Thomas Ntounis, Vasilios Pergialiotis, Ekaterini Domali, Stavros Athanasiou, George Daskalakis, Alexandros Rodolakis, Periklis Panagopoulos, Kalliopi I. Pappa

**Affiliations:** 11st Department of Obstetrics and Gynecology, Alexandra Hospital, National and Kapodistrian University of Athens, 11527 Athens, Greece; 23rd Department of Obstetrics and Gynecology, Attikon Hospital, National and Kapodistrian University of Athens, 11527 Athens, Greece

**Keywords:** sFLT1, PlGF, sFLT1/PlGF ratio, twin pregnancy, twins, adverse outcomes

## Abstract

Twin pregnancies demonstrate a 2–3-fold higher chance of developing PE compared to singletons, and recent evidence has demonstrated that the sFLT1/PIGF ratio is strongly associated with PE, adverse pregnancy outcomes, as well as imminent deliveries due to PE complications. The primary objective of this systematic review was to summarise the available data on the levels of sFLT1, PlGF and their ratios in twin pregnancies and to investigate their association with the development of PE, adverse pregnancy outcomes and the timing of the delivery. A systematic search of Ovid Embase, Web of Science, Science Direct, PubMed, Ovid Medline, Google Scholar and CINAHL was carried out. sFLT1 levels and the sFLT1/PIGF ratio appeared higher in twins compared to singleton pregnancies, especially in the third trimester, while PlGF levels appeared higher up until the third trimester, with their values showing no difference or being even lower than in singletons thereafter. The sFLT1/PIGF ratio has been reported to be an independent marker of adverse outcomes related to pre-eclampsia and is associated with the mean time until delivery in an inverse manner. Further research is required in order to establish the optimal sFLT1/PIGF cut-off values and to stratify the risk of adverse outcomes in twin pregnancies.

## 1. Introduction

Pre-eclampsia (PE) is a multiorgan pregnancy-specific disorder with an overall incidence of 5–8% among all pregnancies and is one of the main causes of maternal morbidity and mortality globally [1,2]. Twin pregnancies demonstrate a 2–3-fold higher chance of developing PE compared to singletons. Several reasons for such a higher incidence have been suggested, such as higher maternal age, the use of assisted reproduction technology (ART) and the larger placental mass; however, the underlying mechanisms for such a finding remain unclear [3,4,5].

PE has been associated with an underlying dysregulation of angiogenic factors, such as soluble fms-like tyrosine kinase-1 (sFLT1) and placental growth factor (PlGF), and recent evidence has underlined those findings by demonstrating that the sFLT1/PIGF ratio is strongly associated with PE, adverse pregnancy outcomes, as well as imminent deliveries due to PE complications in singleton pregnancies [6,7,8]. The importance of these findings and their reproducibility in a series of research projects has led to several courses of action. First, a number of nations have recommended in their guidelines to use these findings in the management of pregnancies with suspected PE, and second, several investigators have redefined PE according to the levels of those factors, given their association with adverse pregnancy outcomes [9,10].

However, the current evidence regarding twin pregnancies is quite scarce and conflicting so far. The present study will aim to review the available published data that focus on the alterations of sFLT1, PlGF and their ratio in twin pregnancies compared to singleton pregnancies and/or their association with the time until delivery and the development of PE and adverse pregnancy outcomes.

## 2. Materials and Methods

### 2.1. Eligibility Criteria, Information Sources, Search Strategy

The literature search was carried out by the two reviewers, I.S. and A.R., across seven databases: PubMed, OVID MEDLINE, Google Scholar, OVID EMBASE, Web of Science, Science Direct and CENTRAL (Cochrane Central Register of Controlled Trials) between 19 November and 20 November 2022. The PRISMA guidelines were followed for the drafting of this paper. Each reviewer used the same search strategy comprising all the necessary keywords (sFLT1, PlGF, sFLT1/PlGF ratio, pre-eclampsia, twin pregnancies, twins, adverse outcomes) and a period spanning from January 2010 to September 2022 in all databases. CENTRAL was searched by both reviewers. The abstracts of the citations were examined by I.S. and A.R. to identify all potentially relevant articles, which were then examined in full-text form. Reference lists of relevant original and review articles were hand-searched for additional reports. Email alerts were put in place to notify the reviewers of new results after the search period. The two reviewers screened the search results independently, and discussions were held at both screening stages alongside a third author (V.P.) to resolve any disagreements before the final papers were decided.

### 2.2. Study Selection

Selection criteria were based on the predetermined characteristics, starting with the population in question being pregnant women of any age with twin pregnancies in which the angiogenic factors in question were examined. It should be noted that we included all the available research that assessed sFLT1, PlGF and their ratio, irrespective of the gestational age of the laboratory assessment. 

Furthermore, we ended up including only the projects that assessed, first, the levels and values of those factors in twin pregnancies and, second, their potential association with pre-eclampsia and composite adverse pregnancy outcomes (one of the following: iatrogenic delivery for hypertensive complications of pregnancy, small-for-gestational-age birth weight).

### 2.3. Data Extraction

The search identified 199 potentially relevant studies, however, 186 were excluded because they were non-relevant articles (abstracts, letters, expert opinions). In total, only 13 peer-reviewed papers were considered to be relevant, and their data was extracted [11,12,13,14,15,16,17,18,19,20,21,22,23]. In three out of the 13 included studies, the populations consisted of combined singleton and twin pregnancies [17,18,22], whereas, in the remaining nine, only twin pregnancies were included [11,12,13,14,15,16,19,20,21]. Two of the included studies focused only on the levels of sFLT1, PlGF and their ratio in twin pregnancies [21,22]; another one focused on the values of both of those angiogenic factors and their association with PE and adverse pregnancy outcomes [13], whereas the rest aimed at examining the potential values of those placental factors in the development of PE and other adverse pregnancy outcomes [11,12,14,15,16,17,18,19,20]. The search strategy is depicted in Figure 1. The main points of the studies finally used can be seen in Table 1, and the summary of the demographic characteristics—wherever those were provided—can be found in Table 2.Two studies [18,23] categorised the demographic characteristics of the patients included according to the outcome and not according to the type of pregnancy, and, as such, they were excluded from the table. 

Given the heterogenicity of the included studies in terms of the populations that were examined (four studies combined singleton and twin populations, while seven included only twin pregnancies), the diversity of the outcome measures (three studies assessed the association of the investigated biomarker with the mean time until delivery, and seven studies investigated their association with PE and only two with selective intrauterine growth restriction) and the different gestational periods of investigation for the biomarkers under question, a meta-analysis was deferred.

### 2.4. Assessment of Risk of Bias

The methodological qualities of the included studies were assessed by two independent reviewers using the QUADAS-2 (Quality Assessment of Diagnostic Accuracy Studies-2) tool that evaluates the technique of patient selection, the indexed test, the reference standard to that test and the flow and timing of the test/study. When the two authors disagreed, a final consensus was given by a third reviewer (V.P.) (Figure 2). The overall risk was decided based on the suggestions made by the QUADAS-2 group [24].

## 3. Results

### 3.1. sFLT1, PlGF, sFLT1/PlGF Ratio Alterations in Twin Pregnancies

Faupel-Badger et al. assessed the changes in sFLT1, PlGF and the sFLT1/PlGF ratio in multiple pregnancies compared to singletons longitudinally through pregnancy [22]. The researchers included data from the BIRTH cohort [25] and a twin study from the Geisel School of Medicine [26], including a total of 132 twins and 2255 singleton controls. Blood sampling took place at four different time points during pregnancy in the BIRTH cohort (median weeks of gestation: 9.7, 17.8, 25.9 and 35.1), while the angiogenic factors were assessed in the third trimester and during labour in the Geisel medical school study. In both cohorts, it was demonstrated that sFLT1, as well as the ratio of sFLT1/PlGF, were higher in twins compared to singletons, while PlGF showed increased values in multiple pregnancies until mid-trimester, followed by lower levels compared to singletons in the third trimester. In terms of the dynamics of those angiogenic factors, sFLT1 showed an increase throughout the pregnancy, and PlGF values increased up until mid-pregnancy, followed by a decline in the third trimester. As expected, their ratio showed a decrease through mid-pregnancy and an increase in the third trimester.

Similar results were extracted by a recent study [13], which included 269 twins, a sample that was derived from the PROGNOSIS, the STEPS and a case-control study [27,28,29]. In this study, seven gestational points were assessed (10 to 14^+6^ w, 15 to 19^+6^ w, 20 to 23^+6^ w, 24 to 28^+6^ w, 29 to 33^+6^ w, 34 to 36^+6^ w, 37 w to delivery), and it was shown that, while the median sFLT1/PlGF ratio demonstrated no difference between multiple and singleton pregnancies up to 29 weeks of gestation, its levels appeared higher thereafter. sFLT1 values appeared higher through every gestational age window, with the rise being more prominent after 29 weeks of gestation, a fact that is in accordance with the increased sFLT1/PlGF ratio levels during the third trimester. In terms of the PlGF levels, their concentrations appeared higher in general except in early pregnancy (10 to 14^+6^ weeks) and between 29 to 37 weeks, with a sharp increase beyond that gestational age (37 weeks to delivery).

Recently, Kozlowski et al. published the first study to associate the concentrations of the biomarkers under question with chorionicity. [21] It was a prospective observational study that included 79 twins—43 monochorionic (MC) and 36 dichorionic (DC)—that were followed up through their pregnancy with two blood samplings, one in the first (11^+0^ to 13^+6^ weeks of gestation) and one in the third trimester (32^+0^ to 34^+0^ weeks of gestation). It was demonstrated that, in the first trimester, sFLT1 was significantly higher in DC compared to MC, while PlGF and the sFLT1/PlGF ratio showed no statistically significant difference among the two groups. sFLT1 remained higher in DC compared to MC twin pregnancies in the third trimester, PlGF was still no different among the two groups, however, the sFLT1/PlGF ratio appeared significantly higher in the DC group due to the higher rise in sFLT1. Finally, in accordance with the previously published studies, it was shown that, in both DC and MC pregnancies, the angiogenic factors under examination showed an increase in the third trimester compared to the first, with the PlGF rise being the most prominent of alterations that led, as expected, to lower levels of the sFLT1/PlGF ratio.

### 3.2. Mean Time until Delivery (MTUD)

Rana et al. [19] was the first to report an inverse correlation between the sFlt/PlGF ratio and the duration of pregnancy (r = −0.25, *p* = 0.03), with the correlation being stronger for gestational periods of less than 34 weeks (r = −0.36, *p* = 0.03).

In their twin study, Binder et al. [15] investigated the association of the sFLT1/PlGF ratio with delivery within 1 or 2 weeks due to PE. According to their findings, the sFLT1/PlGF ratio was significantly higher in women who delivered within 1 and 2 weeks compared with those who did not (within 1-week median: 98.9 versus 27.6; *p* < 0.001, within 2-weeks median: 84.2 versus 23.5; *p* < 0.001), while the levels of PlGF were significantly lower in both study samples compared to women that remained pregnant after 2 weeks. They concluded that a cut-off of 38 could rule out delivery within 1 and 2 weeks with a negative predictive value of 98.8 (95% CI, 92.4–99.8%) and 96.4% (95% CI, 90.1–98.8%), respectively.

Similarly, Karger et al. [15] used the sFLT1/PlGF ratio cut-off of 53, proposed by Droge et al. [17], to diagnose both early and late-onset PE and to assess its value in estimating MTUD. They demonstrated that in early-onset PE (<34 weeks of gestation), MTUD was significantly prolonged in women with a normal ratio compared to the group with a significantly elevated sFLT1/PIGF ratio (≤53 vs. >53: 905.23 h ± 643.08 vs. 220.90 h ± 217.65, *p* = 0.010), a finding that could not be reproduced in late-onset PE cases.

### 3.3. Pre-Eclampsia and Adverse Outcomes

The first study in this review to evaluate the role of angiogenic factors in women with twin pregnancies and suspected pre-eclampsia was conducted by Rana et al. [19]. In this prospective study, researchers enrolled 79 women with twin pregnancies and suspected pre-eclampsia. Blood sample collection (sFLT1, PlGF) occurred at the initial visit in the 3rd trimester, and the outcomes were obtained 2 weeks later. A comparison of 52 (65.8%) pregnancies with pre-eclampsia-related adverse outcomes to 27 (34.2%) uneventful pregnancies revealed that the median sFLT1 was elevated (11461.5 pg/mL (8794.0–14,847.5) versus 7495.0 pg/mL (3498.0–10,482.0, *p* = 0.0004), PlGF was lower (162.5 pg/ ml (98.0–226.5) versus 224.0 pg/mL (156.0–449.0), *p* = 0.005) and the sFLT11/PlGF ratio was elevated (74.2 (43.5–110.5) versus 36.2 (7.1–71.3), *p* = 0.0005). The difference in the sFLT1/PlGF ratio was more pronounced when patients presented <34 weeks (97.7 (76.6–178.1) versus 31.7 (6.5–48.7); *p* = 0.001).

Similar results were reproduced by Droge et al. [17]. In their multi-centre case-control study, women with twin pregnancies and PE had more than four times higher median serum levels of sFLT1 compared to those twin gestations without PE. Conversely, serum levels of PlGF were significantly reduced in twin gestations with PE compared to that of the controls. Interestingly, the sFLT1/PlGF ratio in twin gestations was significantly increased not only in cases of severe PE but also in mild PE cases when compared to uneventful twin gestations (168.67 ± 39.95 and 145.58 ± 39.19 vs. 13.29 ± 319.64 (*p* ≤ 0.001 and *p* = 0.037, respectively)).

Shinohara et al. [11] measured the angiogenic markers at 28 + 0 to 30 + 6 weeks of gestation in 78 women with twin pregnancies. Following an ROC analysis, the authors suggested an sFLT1/PlGF cut-off of 22.2 to predict the PE onset within 4 weeks. This cut-off value provided a sensitivity of 100.0%, a specificity of 88.2%, a positive predictive value of 58.8%, and a negative predictive value of 100.0% (AUC: 0.94).

Similarly, Martínez-Varea et al. [12] studied 108 twin pregnancies and demonstrated that women with twin pregnancies and an sFLT1/PlGF ratio ≥ 17 at 24 weeks had a significantly increased frequency of developing pre-eclampsia (odds ratio, 37.13 (95% CI, 4.78–288.25); *p* = 0.002) and foetal growth restriction (FGR) (odds ratio, 39.58 (95% CI, 6.31–248.17); *p* < 0.001). The mean sFLT1/PlGF ratio, though, was not significantly different amongst the two groups (29.8 PE vs. 18.45 FGR, *p* = 0.42).

On the contrary, Saleh et al. [16] evaluated sFLT1, PlGF and their ratio in singleton versus twin pregnancies and did not report significant differences in biomarkers between PE and non-PE twin pregnancies. However, when the authors compared singleton pre-eclamptic pregnancies to pre-eclamptic twin pregnancies, the levels of sFLT1 (9134 vs. 8625 pg/mL) did not differ, while values of PlGF (185 vs. 33 pg/mL, *p* < 0.001) were increased and values of the ratio (49 vs. 158, *p* = 0.002) were decreased in twin pregnancies with pre-eclampsia.

More recently, Karge et al. [14] did not observe significant differences in the median sFLT1/PlGF ratio between twin pregnancies with suspected PE/HELLP and those with proven PE/HELLP (69.80, IQR: 34.60–97.70 vs. 49.50, IQR: 12.00–74.00; *p* = 0.075). They also conducted an ROC analysis that revealed no predictive value for sFLT1/PIGF and composite perinatal adverse outcome (AUC = 0.618, 95% CI: 0.387–0.849, *p* = 0.254), however, it demonstrated a significant association with sFLT1/PIGF and s-FGR (AUC = 0.755, 95% CI: 0.545–0.965, *p* = 0.032).

Finally, expressed as adjusted multiples of the median, Boucoiran et al. [18] investigated the imbalance of biomarkers in two visits (visit 1: 12–18 weeks; visit 2: 24–26 weeks) in 772 patients. For the 74 multiple-gestation pregnancies, among the biomarkers, PlGF at visit 1 showed the best predictive accuracy for both pre-eclampsia and smallness for gestational age (SGA) (at a 10% false positive rate: 60% and 27% sensitivity for the prediction of PE and SGA, respectively).

## 4. Discussion

### 4.1. Main Findings and General Discussion

The present work aims to summarise the currently published data on the alterations in the levels of the angiogenic factors sFLT1 and PlGF and their ratio during pregnancy, as well as their association with adverse pregnancy outcomes. A summary of the observations and main conclusions of the included studies is depicted in Table 3.

According to the available evidence, sFLT1 levels appeared higher in twin pregnancies throughout the pregnancy, generating a massive rise, especially after 29 weeks of gestation, while PlGF levels began by being slightly higher only to decrease in no different or even lower levels compared to singletons in the third trimester. [13,22] Contrary to the above, Kozlowski et al. [21] demonstrated a more prominent rise in PlGF (10-fold increase in MC twins) in the third trimester compared to sFLT1(2-fold increase in MC twins). The sFLT1 rise was initially attributed to the increased placental volume in twin pregnancies, especially at the end of the pregnancy, given the fact that the placenta is responsible for its production. However, the investigated association between placental weight and the aforementioned angiogenic factors showed, first, no correlations with sFLT1 and, second, only a weak positive correlation with PlGF [22]. As such, other potential underlying mechanisms could co-exist, such as ongoing endothelial damage of the spiral arteries that take place in twin pregnancies altering the angiogenic profile of those patients or maternal factors such as age and race.

In terms of the effect of the underlying chorionicity on the above-mentioned factors, Binder et al. [15] and Calle et al. [13] demonstrated no differences between MC and DC pregnancies, while Faupel-Badger et al. concluded that MC twins have higher values of sFLT1 and sFLT1/PlGF ratios. The results were, however, extracted from a small sample size (n = 5 MC, n = 36 DC). The available evidence becomes even more confusing after the results of Kozlowski et al., which, after including a bigger sample size (n = 43 MC, n= 36 DC), revealed an increased concentration of sFLT1 in DC pregnancies in both the first and third trimesters, the while PlGF levels did not appear to be different between MC and DC twins in any of the study periods. The subsequent higher levels of the sFLT1/PlGF ratio in DC pregnancies may explain the higher incidence of PE noted in DC twins. The observed findings were attributed, but not limited to, a combination of increased placental weight and placental ischemia by the authors.

As stated earlier, an elevated sFLT1/PlGF ratio is associated with shorter MTUD. Several studies have assessed such a correlation in twin pregnancies, reproducing similar outcomes as in singletons. Rana et al. [19] demonstrated that high ratio levels could lead to an iatrogenic delivery within 2 weeks. Karge et al. [14] showed that levels above the cut-off of 53 were correlated with a shorter time until delivery. Binder et al. even used the ratio as a negative predictive tool for delivery because of early-onset PE within 2 weeks (suggesting a cut-off value of 38) [15]. All authors acknowledged a series of limitations in their studies, such as their non-blinded design, the possibility of intervention bias and their retrospective nature. Nevertheless, they concluded that the implementation of the assessment of the sFlt-PlGF ratio in cases with suspected PE/HELLP might intensify the follow-up of those pregnancies with a subsequent potentially beneficial effect.

The role of angiogenic biomarkers in singleton pregnancies with PE has been well established [30]. Current data regarding their efficacy when managing twin pregnancies with PE are limited and conflicting. Rana et al. [19] reported that the sFLT1/PlGF ratio is higher (74.2 versus 36.2) in twin pregnancies with adverse outcomes compared to those without and noted that their findings were similar to singleton pregnancies. This difference was more prominent in patients before 34 weeks (97.7 versus 31.7). Similar results were observed by Binder et al. [15] in women requiring delivery because of pre-eclampsia within 1 week of blood sampling compared with those who did not deliver (sFLT1/PlGF ratio: 98.9 versus 27.6). Interestingly, both studies found that the sFLT1/PlGF ratio is an independent marker of adverse outcomes regardless of the gestational age at sampling. This is an important implication for clinical practice, enabling the diagnosis and risk stratification of patients with twin pregnancies at high risk for PE and PE-related adverse maternal and perinatal outcomes. This ratio could be a useful diagnostic tool, even in situations where accurate gestational age information is not available because of insufficient prenatal care and no previous ultrasound assessment.

Several studies have suggested different levels of angiogenic markers in singleton and twin pregnancies, highlighting that those cut-offs used in singleton pregnancies for the prediction of PE and/or adverse outcomes related to PE may not be applicable in twin pregnancies. Saleh et al. [16] and Droge et al. [17] demonstrated that the use of the sFLT1/PlGF cut-off of 38 suggested for singletons is not appropriate in twin pregnancies for the prediction of pre-eclampsia. However, Saleh et al. [16] was a small cohort of 21 twins, and Droge et al. [17] included extremely high sFLT1/PlGF ratios of 10,000 in their control group that could have biased their cut-off values. Another study by Shinohara et al. [11] suggested that a lower sFLT1/PlGF cut-off of 22.2 measured at 28 ^+0^–30^+6^ weeks of gestation could be a useful predictor for pre-eclampsia in twin pregnancies within 4 weeks from blood sampling. In line with the above findings, however, in earlier gestation, Martínez-Varea et al. [12] observed that an sFLT1/PlGF ratio ≥ 17 at 24 weeks in twin pregnancies was associated with a significant increase in the frequency of pre-eclampsia and FGR. Some possible explanations for the lower cut-off values observed in twin pregnancies with PE compared to singleton counterparts include the increased placental mass and subsequent changes in the release of circulating angiogenic biomarkers, maternal blood volume alteration, as well as the combination of higher maternal cardiac output and lower peripheral resistance in twin pregnancies compared to singletons [31,32].

Although PE and FGR are different entities, they share common pathogenic pathways that include inadequate placentation, inflammation, and maternal vascular dysfunction [33]. Two studies [12,14] in our review reported a significant association between an elevated sFLT1/PlGF ratio and FGR, however, in both studies, the size of the FGR subgroup was quite small.

It is noteworthy that most of these studies are observational studies, and the actual effect of the sFLT1/PlGF ratio assessment when managing twin pregnancies and suspected pre-eclampsia remains unknown.

### 4.2. Strengths and Limitations

To our knowledge, this is the very first systematic review to include all the available published data regarding the association of the angiogenic factors sFLT1, PlGF and the sFLT1/PlGF ratio in twin pregnancies with the subsequent development of PE/HELLP, FGR and the timing of delivery. This review also discusses the alterations of those biomarkers in twins compared to singleton pregnancies.

The main limitation of this review can be summarized as the small sample size of the included studies. To be more precise, eight of the included studies analysed data from less than 100 twin pregnancies, which was emphasised by their respective authors. Furthermore, the methodological study design and the sample population of the included studies were quite heterogenous, given the fact that they were both prospective and retrospective cohort studies as well as case-control studies, while in terms of the populations under examination, four studies mixed singleton and twin populations, and seven only included twin pregnancies. Last but not least, the investigation of the biomarkers under question took place in several gestational periods, ranging from 9 up to 39 weeks of gestation, and, as such, the extraction of any conclusions in terms of the prediction or management of adverse pregnancy outcomes should be guarded. As stated earlier, the above-stated heterogeneities and the diversity of the outcome measures under investigation among the included studies were the reasons that led us to defer conducting a meta-analysis since we concluded that the results would be quite questionable and possibly confusing.

## 5. Conclusions

The alterations of the levels of the angiogenic factors sFLT1, PlGF and their ratio in twin pregnancies are parameters that demonstrate a potential additive value for the prediction of the timing of delivery and the development of PE and foetal growth restriction. Given the low prevalence of twin pregnancies, future research should focus on the design of multicentre studies that would include an adequate sample size of twin pregnancies, which will facilitate concrete results to be extracted. Similar to singleton pregnancies, the aforementioned biomarkers could be implemented in screening and management policies, however, for such strategies to be achieved, the evaluation of those factors needs to be performed in several gestational periods, in different types of twin pregnancies (monochorionic and dichorionic), for the determination of an optimal cut-off and for their potential benefit in terms of perinatal and neonatal outcomes.

## Figures and Tables

**Figure 1 medicina-59-01232-f001:**
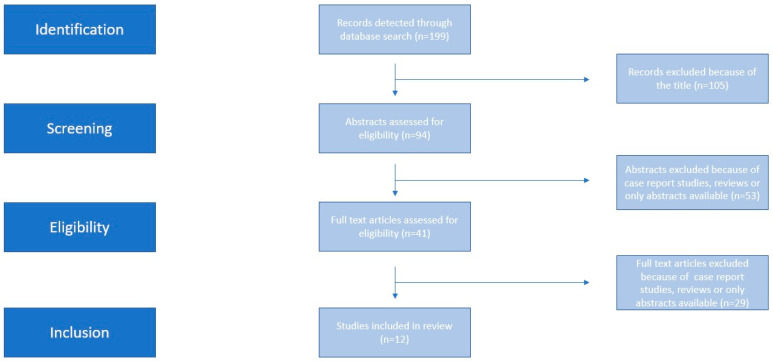
Search strategy.

**Figure 2 medicina-59-01232-f002:**
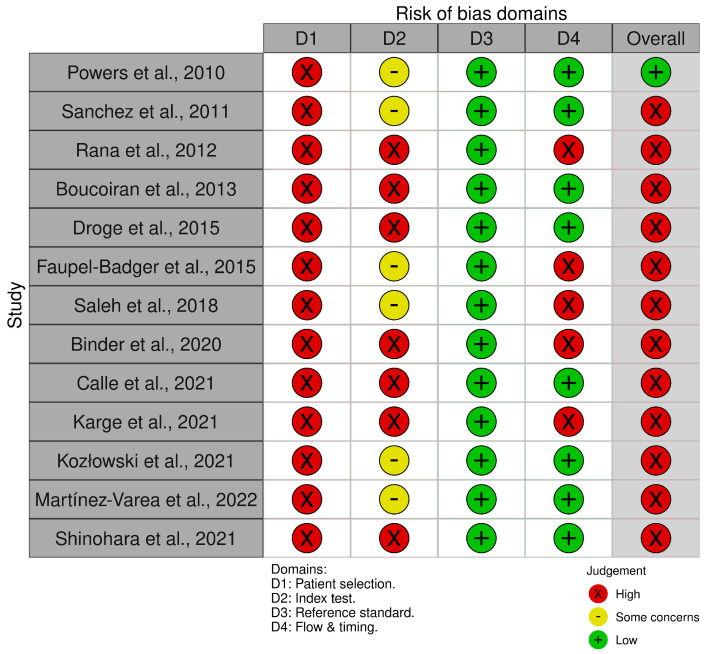
QUADAS-2 assessment [11,12,13,14,15,16,17,18,19,20,21,22,23].

**Table 1 medicina-59-01232-t001:** Summary of the studies included. Abbreviations: PET: Pre-eclampsia; FGR: Fetal Growth Restriction; APO: Adverse Pregnancy Outcome; AUC: Area Under the Curve; GA: Gestational Age; PPV: Positive Predictive Value; NPV: Negative Predictive Value.

Authors	Type of Study	Mean GA at Blood Sampling in Weeks of Gestation (Range)	Inclusion	N (Twins)	Control	Outcome	Pregnancy Duration from Presentation to Delivery	sFLT1/PlGF Ratio	sFLT1 (pg/mL)	PIGF (pg/mL)
Powers et al., 2010 [20]	Secondary analysis of a multicentre randomised controlled trial	Visit 1: 7–26 Visit 2: 24–28 Visit 3: 34–38	Multifetal gestations	39 with PET	195 without PET	Prediction of PET in high-risk pregnancies	N/A	N/A	With PET 7330 ± 5420 Without PET 5950 ± 2470	With PET 386.1 ± 318.17 Without PET 554.73 ± 388
Rana et al., 2012 [19]	Prospective Cohort	33.9 (31.9–36.0)	Twins with suspected PET	52 with adverse outcome 27 without adverse outcome	N/A	**(1)** Adverse outcomes **(2)** Time until delivery	**With adverse** 3.5 days **Without adverse** 14.5 days	**With adverse** 74.2 **Without adverse** 36.2	**With adverse**: 11,461.5 **Without adverse adverse**: 7495.0	**With adverse**: 162.5 **Without adverse adverse:** 224.0
Boucoiran et al., 2013 [18]	Prospective Cohort	**Visit 1** 12–18 **Visit 2** 24–26	Twins and singletons	69	703	Predictive accuracy of PlGF, sFLT1, and inhibin A plasma concentrations in multiple compared to singleton pregnancies	N/A	**Visit 1 ** With PET 2.53 (0.96–3.45) MoM Without PET 1.09 (0.55–1.71) MoM **Visit 2 ** With PET 21.23 (0.55–34.31) MoM Without PET 0.73 (0.42–1.74) MoM	Visit 1 With PET 0.72 (0.70–0.87) MoM Without PET 1.06 (0.73–1.56) MoM **Visit 2 ** With PET 2.21 (0.99–3.87) MoM Without PET 1.03 (0.55–1.65) MoM	**Visit 1 ** With PET 0.34 (0.31–0.49) MoM Without PET 1.08 (0.74–1.47) MoM **Visit 2 ** With PET 0.22 (0.10–1.03) MoM Without PET 1.08 (0.54–1.78) MoM
Droge et al., 2015 [17]	Multicenter case–control study	No PE twins: 30.5 PE twins: 32.9	Twins and singletons	18 with PET 31 without PET	54 singletons with PET 238 singletons without PET	PET	N/A	**With PET** 164.22 **Without PET** 13.29	**With PET** 20,011.50 pg/mL **Without PET** 4503.00 pg/mL	**With PET** 138.80 pg/mL **Without PET** 403.00 pg/mL
Faupel-Badger et al., 2015 [22]	Data analysis from two studies	**BIRTH**: 4 visits 9.7 (8.4–11.6), 17.8 (16.8–18.7), 25.9 (24.8–28.1), and 35.1 (34.6–35.9) **Geisel School of Medicine:** 31–39 weeks and after admission for labor	Twins and signletons without PET	**BIRTH** 91 **Geisel School of Medicine** 41	**BIRTH** 2193 **Geisel School of Medicine** 62	Comparison of angiogenic factors between twins and singletons	N/A	**BIRTH (Twins vs. Singletons)** 9.7 weeks: 294.6 vs. 202.4 17.8 weeks: 58.7 vs. 44.3 25.9 weeks: 19.4 vs. 13.2 35.1 weeks: 168.4 vs. 29 **Geisel School of Medicine(Twins vs. Singletons)** 3rd trimester:15.9 vs. 4.51 Delivery: 107.8 vs. 47.8	**BIRTH (Twins vs. Singletons)** 9.7 weeks: 7037 vs. 4485 17.8 weeks: 12,543 vs. 6131 25.9 weeks: 12,968 vs. 5898 35.1 weeks: 36,916 vs. 10,151 **Geisel School of Medicine(Twins vs. Singletons)** 3rd trimester: 6129 vs. 2108 Delivery: 15,899 vs. 7278	**BIRTH (Twins vs. Singletons)** 9.7 weeks: 24.9 vs. 22.8 17.8 weeks: 213.5 vs. 138.4 25.9 weeks: 668 vs. 445.9 35.1 weeks: 219.2 vs. 350.2 **Geisel School of Medicine (Twins vs. Singletons)** 3rd trimester: 386.2 vs. 467.3 Delivery: 147.5 vs. 152.4
Saleh et al., 2018 [16]	Secondary analysis of a prospective multicenter cohort study	29 (23–34) 30 (24–34)	Twins with suspected PET	21	21 singletons	**(1)** PET **(2)** Adverse pregnancy outcomes	N/A	**Confirmed PET twins**: 49 **Suspected PET twins**: 26	**Confirmed PET twins:** 9134 **Suspected PET twins:** 6377	**Confirmed PET twins:** 185 **Suspected PET twins:** 228
Binder et al., 2020 [15]	Retrospective analysis	33.6 (30.0–35.2)	Twins with suspected PET	164	N/A	Delivery because of PET within 1 or 2 weeks of blood sampling	N/A	**≤1 wk due to** **PET**: 98.9 **≤2 wk due to** **PET**: 84.2 **Delivery >2 wk due to** **PET or at Any** **Time Reasons Other** **Than PET**: 23.5	**≤1 wk due to** **PET**: 15,034 **≤2 wk due to PET:** 14,620 **Delivery > 2 wk due to** **PET or at Any** **Time Reasons Other** **Than PET:** 6954	**≤1 wk due to** **PET**: 150.2 **≤2 wk due to** **PET**: 62.1 **Delivery > 2 wk due to** **PET or at Any** **Time Reasons Other** **Than PET**: 344.8
Calle et al., 2021 [13]	Reference range analysis	**PROGNOSIS:** 26–37 **STEPS: Visit 1:** 19 or 20, **Visit 2:** 23–24, **Visit 3**: 27–28	Twins	269	N/A	**(1)** Reference ranges for sFLT1/PlGF ratio in twin pregnancies **(2)** Predictive performance short-term PE	N/A	Cut-off of 38 NPV of 91.9 and 83.8% to rule out PE within 1 and 4 weeks	N/A	N/A
Karge et al., 2021 [14]	Retrospective cohort study	N/A	Twins with suspected PET and/or HELLP syndrome	49	N/A	**(1)** Adverse perinatal outcome. **(2)** Mean time until delivery	**< 34 weeks** **sFLT1/PIGF ratio ≤ 53**: 905.23 h ± 643.08 **sFLT1/PIGF ratio > 53**: 220.90 h ± 217.65 **sFLT1/PIGF ratio ≤ 85**: 741.48 h ± 624.94 **sFLT1/PIGF ratio > 85**: 109.00 h ± 119.03 **≥34 weeks** **sFLT1/PIGF ratio ≤ 53**: 113.70 h ± 157.03 **sFLT1/PIGF ratio > 53**: 123.03 h ±157.03 **sFLT1/PIGF ratio ≤ 110**: 127.42 h ± 144.88 **sFLT1/PIGF ratio > 110**: 74.67 h ± 82.16	**With suspected PE ** 69.80 **With PE ** 49.50 **With adverse** 89.45 **Without adverse** 62.00	N/A	N/A
Kozłowski et al., 2021 [21]	Prospective observational study	**Visit 1:** 11–14 **Visit 2:** 32–34	Twins	79	N/A	Expression of angiogenic biomarkers in dichorionic and monochorionic twins	N/A	N/A	N/A	N/A
Shinohara et al., 2021 [11]	Retrospective observational cohort	29 (28–30)	Twins	10 with PET within 4 wks	68 without PET within 4 weeks	Development of PET within 4 weeks	**With PET** 4.7 weeks **Without PET** 7.3 weeks	**With PET**: 46.2 (22.2–64.6) **Without PET**: 4.0 (0.5–70.7)	N/A	N/A
Martínez-Varea et al., 2022 [12]	Prospective study	24	Twins	14 with PET/FGR	94 without PET/FGR	Prediction of PET/FGR	N/A	**With PET/FGR** 20.286 (22.317) **Without PET/FGR ** 4.309 (7.008)	N/A	N/A

**Table 2 medicina-59-01232-t002:** Summary of the demographic characteristics of the studies included. Abbreviations: wk: week; N/A: non applicable; GDM: gestational diabetes; PET: pre-eclampsia; HELLP: Haemolysis Elevated Liver Enzymes and Low Platelets; * *p* ≤ 0.05; ** *p* ≤ 0.001.

Demographics	Powers et al., 2010 [20]	Rana et al., 2012 [19]	Boucoiran et al., 2013 [18]	Droge et al., 2015 [17]	Faupel-Badger et al., 2015 [22]		Saleh et al., 2018 [16]	Binder et al., 2020 [15]		Calle et al., 2021 [13]	Karge et al., 2021 [14]		Kozłowski et al., 2021 [21]	Shinohara et al., 2021 [11]	Martínez-Varea et al., 2022 [12]
**N (total)**	234 multifetal gestations	**79 twins with suspected PET**	**772 twins and singletons**	341 twins and singletons	**BIRTH** 2284 twins and signletons without PET	**Geisel School of Medicine** 103 twins and singletons without PET	42 twins with suspected PET	164 twins with suspected PET		269 twins	49 twins with suspected PET and/or HELLP syndrome		79 twins	78 twins	108 twins
n	39 with PET	52 twins with adverse outcome	69	18 twins with PET	91 twins	41 twins	13 twins with confirmed PET	**Delivery ≤ 1 wk due to PET** 29	**Delivery ≤ 2 wk due to PET** 42	**PROGNOSIS** 22 **STEPS** 222 **Case-control study of the Elecsys** 25	**Early onset PET/HELLP** 18	**Late onset PET/HELLP** 31	43 monochorionic pregnancies 36 dichorionic pregnancies	10 twins with PET within 4 wks	14 twins with PET and/or FGR
Group of control	195 without PET	27 twins without adverse outcome	N/A	31 twins without PET 54 singletons with PET 238 singleton without PET	2193 singletons	62 singletons	8 twins with suspected PET 6 singletons with suspected PET 15 singletons with confirmed PET	**Delivery > 2 wk due to PET or at Any Time Reasons Other Than PET** 122		N/A	N/A	N/A	N/A	68 twins without PET within 4 wk	94 twins without PE or FGR
Maternal age (years)	26 ± 7	34.0 (32.0–38.0) *	N/A	33.56 ± 5.35 **	35.1 (5.8) **	33.4 (5.9) *	36 (31–44)	37.0 (33.0–39.0) *	36.0 (31.2–38.0)	34 (mean)	34.61 ± 4.67	33.74 ± 5.59	31.5 ± 4.3	34 (27–43)	36.000 (4.930)
Nulliparous	19 *	43/52 *	N/A	N/A	27/91	12/41	N/A	16 /29	25/42	N/A	15/18	21/31	39/79	6/10	11 (78.6)
BMI (kg/m2)	29 ± 8	21.9 (27.8–35.2)	N/A	24.09 ± 4.97	N/A	N/A	N/A	24.5 (21.8–26.0)	25.1 (21.9–29.1)	25 (mean)	24.80 ± 5.11	23.94 ± 6.57	23.8 ± 4.6	19.7 (16.9–28.2)	24.192 (3.241)
**Medical History**			N/A												
Previous PE	N/A	1/52 *	N/A	N/A	N/A	N/A	N/A	N/A	N/A	N/A	N/A	N/A	N/A	N/A	0/14
Chronic Hypertension	N/A	7/52	N/A	N/A	N/A	N/A	N/A	N/A	N/A	N/A	14/18	1/31	N/A	N/A	0/14
Renal disease	N/A	1/52	N/A	N/A	N/A	N/A	N/A	N/A	N/A	N/A	N/A	N/A	N/A	N/A	N/A
Diabetes	N/A	1/52	N/A	N/A	N/A	N/A	N/A	N/A	N/A	N/A	1/18 (GDM)	4/31 (GDM)	21/79 (GDM)	3/10 * (GDM)	0/14
Antihypertensive medication	N/A	N/A	N/A	N/A	N/A	N/A	N/A	8/29	13/42 *	N/A	11/18 **	4/31 **	N/A	N/A	N/A
Aspirin during pregnancy	14	N/A	N/A	N/A	N/A	N/A	N/A	N/A	N/A	N/A	5/18	4/31	N/A	N/A	N/A
**Chorionicity/ Method of conception**			N/A												
Dichorionic	N/A	N/A	N/A	N/A	N/A	N/A	N/A	23/29	33/42	N/A	11/18 *	28/31 *	N/A	N/A	9/14
Monochorionic	N/A	N/A	N/A	N/A	N/A	N/A	N/A	6/29	9/42	N/A	7/18	3/31	N/A	N/A	5/14
ART	N/A	N/A	N/A	N/A	64/91 **	16/41 *	10/13	13/29	18/42	N/A	11/18	17/31	17/79	3/10	7/14
**Race**			N/A												
White/Caucasian	N/A	47/52	N/A	14/18 *	64/91	40/41	13/13	N/A	N/A	N/A	17/18	28/31	N/A	N/A	N/A
Black/African American	N/A	1/52	N/A	1/18	9/91	0/41	N/A	N/A	N/A	N/A	N/A	N/A	N/A	N/A	N/A
Asian	N/A	4/52	N/A	0/18	9/91	0/41	N/A	N/A	N/A	N/A	N/A	N/A	N/A	N/A	N/A
Other/Unknown	N/A	0/52	N/A	3/18	2/91	1/41	N/A	N/A	N/A	N/A	N/A	N/A	N/A	N/A	N/A
**Family history of PE**			N/A												
Yes	N/A	N/A	N/A	0/18	N/A	N/A	N/A	N/A	N/A	N/A	N/A	N/A	N/A	3/10	N/A
No	N/A	N/A	N/A	14/18	N/A	N/A	N/A	N/A	N/A	N/A	N/A	N/A	N/A	N/A	N/A
Unknown	N/A	N/A	N/A	4/18	N/A	N/A	N/A	N/A	N/A	N/A	N/A	N/A	N/A	N/A	N/A

**Table 3 medicina-59-01232-t003:** Summary of the observations and main conclusions of the included studies. Abbreviations: wk: week; N/A: non applicable; GDM: gestational diabetes; PE: pre-eclampsia; HELLP: Haemolysis Elevated Liver Enzymes and Low Platelets;UtA: uterine artery Doppler; FGR: Fetal Growth Restriction; MTUD:Mean time until delivery.

Authors	Type of Study	Conclusions/Observations
Powers et al., 2010 [20]	Secondary analysis of a multicentre randomized controlled trial	sFlt1 and PlGF are significantly higher among women with multifetal gestations compared with other high-risk groups Observed that sFlt1 and PlGF modest significant differences of at least one of these factors during the third trimester in women who develop preeclampsia in all high-risk groups including multifetal gestations
Rana et al., 2012 [19]	Prospective Cohort	sFlt1/PlGF ratio at the time of initial evaluation is associated with subsequent adverse maternal and perinatal outcomes in women with twin pregnancy and suspected preeclampsia
Boucoiran et al., 2013 [18]	Prospective Cohort	PlGF level was a good predictor of subsequent PE as early as 12 to 18 weeks in multiple-gestation pregnancies but was not clinically useful enough to be used as a single marker
Droge et al., 2015 [17]	Multicenter case–control study	In twin pregnancies with PE, sFlt-1 levels and the sFlt-1/PlGF ratio were increased and PlGF levels were decreased as compared to twin gestations with an uneventful pregnancy outcome sFlt-1/PlGF ratio did not differ between twin pregnancies with PE and singleton pregnancies with PE. In twin pregnancies with an uneventful outcome, sFlt-1 levels and sFlt-1/PlGF ratio were increased, but no differences in PlGF concentration were found when compared with that of singleton controls. Reference ranges of sFlt-1, PlGF and their ratio in singleton pregnancies are therefore not transferable to twin pregnancies
Faupel-Badger et al., 2015 [22]	Data analysis from two studies	sFlt-1 concentrations and the sFlt-1/PlGF ratio were higher in twins than singletons across pregnancy and at delivery, with the greatest differences at week 35. PlGF concentrations were lower in twin than singleton pregnancies at week 35 only. Placental weight appeared to be inversely correlated with maternal sFlt-1/PlGF ratio at the end of pregnancy in both twins and singletons
Saleh et al., 2018 [16]	Secondary analysis of a prospective multicenter cohort study	Serum sFlt-1levels are considerably higher in twin than in singleton control gestations. sFlt-1/PlGF ratio of ≤38 to predict short-term absence of PE is not applicable to twin pregnancies in predicting either the absence of PE or the absence of adverse pregnancy outcomes
Binder et al., 2020 [15]	Retrospective analysis	sFlt-1/PlGF ratio lower than 38 was able to rule-out delivery within 1 and 2 weeks with a negative predictive value of 98.8% and 96.4% for delivery because of preeclampsia within 1 and 2 weeks, respectively. A cutoff of 38 is applicable for ruling out delivery because of preeclampsia in twin pregnancies
Calle et al., 2021 [13]	Reference range analysis	Up to 28 weeks + 6 days’ gestation, median, 5th, and 95th percentile values for the sFlt-1/PlGF ratio in twin pregnancies were similar to singleton pregnancies. From 29 weeks of gestation onward, median, 5th, and 95th percentile values for the sFlt-1/PlGF ratio appear to be higher in twin pregnancies suggesting that ratio could be useful in those pregnancies
Karge et al., 2021 [14]	Retrospective cohort study	sFlt-1/PlGF ratio in twin pregnancies with suspected PE/HELLP may be useful for the prediction of adverse perinatal outcome, especially to identify cases of s-FGR. MTUD was significantly shortened in women with an elevated sFlt-1/PIGF ratio, an intensified clinical monitoring is required, mainly in women with early onset PE and a ratio > 85. However, a normal ratio may not rule out adverse perinatal outcome.
Kozłowski et al., 2021 [21]	Prospective observational study	sFlt-1 level was related to twin gestation chorionicity (significantly higher concentration of sFlt-1 in dichorionic in comparison to monochorionic pregnancies in both the first and third trimesters)
Shinohara et al., 2021 [11]	Retrospective observational cohort	A cutoff value of 22.2 for the sFlt-1/ PlGF ratio at 28–30 weeks of gestation may be useful to exclude the development of PE within 4 weeks in twin pregnancies
Martínez-Varea et al., 2022 [12]	Prospective study	sFlt-1/PlGF ratio ≥17 at 24 weeks in twin pregnancies is associated with a significant increase in the frequency of preeclampsia and FGR. sFlt-1/PlGF ratio at 24 weeks in twin pregnancies, combined with the mean PI UtA and maternal characteristics, could select patients at risk for placental dysfunction, such as preeclampsia or FGR

## Data Availability

This is a review of the literature, and as such, no new data were created.

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
