# Peer review of "sFLT1, PlGF, the sFLT1/PlGF Ratio and Their Association with Pre-Eclampsia in Twin Pregnancies—A Review of the Literature"

_medicina, 2023, doi:10.3390/medicina59071232_

Round 1

Reviewer 1 Report

The manuscript by Sapantzoglou et al. is a review of sFLT1, PlGF and their ratio in preeclampsia with twin gestations. The authors do a reasonable job of reviewing the literature on this topic, but they are missing some relevant studies that should have been included, and the manner in which the data is displayed is challenging and can be confusing. Additional specific comments include: 

1.     In all instances throughout the manuscript, including the title, sFLT1 should be spelled as such, and not as “sFlt1” or "Sflt1”. These spellings are incorrect or refer to the mouse protein/ gene. 

2.     The authors switch between past and present tense in several parts of the manuscript, including line 13 in the abstract. Please pick a tense and make it consistent throughout the manuscript. 

3.     Lines 42 and 85, instead of “adversities in pregnancy” it would be better and more accurate to write “adverse pregnancy outcomes”. 

4.     Line 45, change sFlit-1 to sFLT1. 

5.     What was the rationale for choosing Jan 2010 as the start date for the literature search?

6.     The authors are missing other relevant papers related to this topic including: PLoS One. 2010 Oct 11;5(10):e13263. doi: 10.1371/journal.pone.0013263.

7.     Why are there no values reported for sFLT1, PlGF and the sFLT1/PlGF ratio in Table 1 for some studies, namely Boucoiran et al., De La Calle et al., Karge et al., and Kozlowski et al.? The cited works appear to contain the data. 

8.     For the assessment of risk bias, how were differences between the two independent reviewers resolved?

9.     How was the final “overall” risk bias for each of the cited papers determined, especially since all received a “high” unsatisfactory rating?

10.  On page 9, line 17, please remove the indication that Faupel-Badger et al. were the “first” to assess sFLT1, PlGF and the ratio in multifetal pregnancies longitudinally. This is incorrect. The authors are missing other studies. 

11.  It would be extremely helpful to the reader to have a summary figure for the summary data reported in the manuscript. Listing all of the various highlights of the data from the various studies is somewhat useful, but also very tedious.

12.  Page 11, line 83, please change “reduced” to “lower”. The term reduced is only accurate in pairwise longitudinal studies where the change in a factor is followed over time. This correction may also be necessary in other parts of the manuscript. 

13.  Page 12, line 128, remove the “first review” label. It is certainly possible there is another review that is “first”, and this kind of description is unnecessary and unhelpful. 

14.  The manuscript is written as a general overview of several relevant and some smaller studies, which begs the question as to whether this would be better as a metanalysis. A metanalysis of all the relevant data combined would make a stronger study. 

15.  Page 13, line 211-214, delete this sentence. 

16.  Page 14, line 234, correct the spelling of sFLT1, not s-FGR (?). 

The quality of the English can be improved. I have provided several examples. 

Author Response

Dear reviewer,

Thank you for your time and the revision you have performed on our manuscript. The following changes have been made:

  1. The term sFLT1 is used throughout the manuscript and the necessary changes have been made.
  2. Past tense was used throughout the abstract for it to be consistent.
  3. The necessary changes were made as was suggested.
  4. The change has been made.
  5. The authors aimed to perform a systematic review of the recent bibliography and as such the past 20 years were included in the search.
  6. The paper mentioned was included in our systematic review with its data and conclusions included in the manuscript ,as well as, the tables.
  7. Data from Boucoiran were expressed in MoMs and they were added, in De La Calle et al. the authors generated sFlt-1 and PlGF concentrations in normal twin pregnancies for the 7 gestational age windows assessed and then used a cut-off of 38 for the ration and the prediction of PET with the observations added, Karge et al.had no values for Sflt1,PlGF separately and in Kozlowski et al.the authors report serum concentrations of PlGF, sFlt-1, sFlt-1/PlGF in monochorionic and dichorionic groups without hypertension-related disorders and that is why values were not included in our table.
  8. When the two authors disagreed, a final consensus was given by a third reviewer (V.P.).This was added in the Assessment of Risk of bias section of the Materials and Methods.
  9. The overall risk was decided based on the suggestions made by the QUADAS-2 group. This was added in the Assessment of Risk of bias section of the Materials and Methods.
  10. The sentence was removed.
  11. A summary table was created that includes the authors, the study design and the main outcomes or observations of each study.
  12. The requested change was made.
  13. The requested change was made.
  14. Given the heterogenicity of the included studies ,unfortunately, a meta-analysis was deferred. A paragraph was included about that in the Materials and Methods section of the manuscript.
  15. The requested change was made.
  16. The requested change was made and s-FGR(selective fetal growth restriction) was changed to fetal growth restriction, to be more clear.

Reviewer 2 Report

The authors dealt with an interesting topic of the association of sFlt-1, PlGF and Sflt-1/PlGF ratio with the onset of preeclampsia in twin-pregnancies. However, as also stated by authors number of studies included in this review is low (only 11). In addition, considering there is a recent review on this topic (Verlohren et al., 2022) there is concern about the novelty of the present manuscript.
Nevertheless, the authors use therm "the need for imminent delivery". I believe this needs to be rephrased as it is not a need for imminent delivery, it is just an imminent delivery, as it is an unplanned consequence of the condition.
Only two authors searched databases whereas there are 13 authors on this paper. They should have included at least one more, due to decision-making.
The quality of Figure1 is low. Referencing should be adjusted. Instead of 12,13,14,15,16,etc, it is better to write 12-16.

Author Response

Dear reviewer,

Thank you for your comments and for the revision you have performed on our manuscript. I would like to start by mentioning that the recent paper by Verloher et al differs from ours given the fact that it is about ‘’Clinical interpretation and implementation of the sFlt-1/PlGF ratio in the prediction, diagnosis and management of preeclampsia’’ while our paper is a review of how sFLT1,PlGF and their ratio changes throughout the pregnancy and the way those biomarkers are associated with pregnancy outcomes focusing on twin pregnancies only.

The changes we have performed on our manuscript, after your kind suggestions, are the following:

1. The requested change regarding the term ''imminent delivery" was made.

2.Αn additional author was included in the search strategy (V.P.) and that was added in the Materials and Methods section of the manuscript.

3. Figure 1 was reuploaded hoping it would be of better resolution now.

4. The requested change regarding the references was made.

Round 2

Reviewer 1 Report

I believe Figure 2 still needs to be updated. 

Author Response

Dear reviewer,

Thank you for your notification. Figure 2 was changed, assuggested, and now it includes the new added paper of our review.

Kind regards,

Ioakeim Sapantzoglou

Author Response

Dear reviewer,

Thank you for the approval of our changes of the manuscript.

Kind regards,

Ioakeim Sapantzoglou